# Overcoming EGFR Resistance in Metastatic Colorectal Cancer Using Vitamin C: A Review

**DOI:** 10.3390/biomedicines11030678

**Published:** 2023-02-23

**Authors:** Ahmad Machmouchi, Laudy Chehade, Sally Temraz, Ali Shamseddine

**Affiliations:** Department of Internal Medicine, Division of Hematology/Oncology, Naef Khaled Basile Cancer Institute—NKBCI, American University of Beirut Medical Center, Beirut 1107 2020, Lebanon

**Keywords:** vitamin C, ascorbic acid, colorectal cancer, EGFR Resistance, KRAS mutation, BRAF mutation

## Abstract

Targeted monoclonal antibody therapy against Epidermal Growth Factor Receptor (EGFR) is a leading treatment modality against metastatic colorectal cancer (mCRC). However, with the emergence of KRAS and BRAF mutations, resistance was inevitable. Cells harboring these mutations overexpress Glucose Transporter 1 (GLUT1) and sodium-dependent vitamin C transporter 2 (SVCT2), which enables intracellular vitamin C transport, leading to reactive oxygen species generation and finally cell death. Therefore, high dose vitamin C is proposed to overcome this resistance. A comprehensive search strategy was adopted using Pubmed and MEDLINE databases (up to 11 August 2022). There are not enough randomized clinical trials to support its use in the clinical management of mCRC, except for a subgroup analysis from a phase III study. High dose vitamin C shows a promising role in overcoming EGFR resistance in mCRC with wild KRAS mutation with resistance to anti-epidermal growth factor inhibitors and in patients with KRAS and BRAF mutations.

## 1. Introduction

Colorectal cancer (CRC), one of the most common malignant neoplasms, is the third leading cause of death among all malignancies. Depending on the type of cancer, gender and location, it is ranked 2nd to 4th in terms of incidence in the world [1]. Many efforts have been made with the aim of finding the optimal treatment plan to improve the prognosis of CRC. Specifically, in RAS, BRAF wild and MSS tumors, cytotoxic/cytostatic chemotherapy (5-FU), Vascular Endothelial Growth Factor Inhibitor (VEGF), and Multi-Kinase targeted agents, in addition to targeted monoclonal antibodies (against Epidermal Growth Factor Receptor (EGFR), have been widely used as the leading treatment modality against metastatic CRC (mCRC) [2,3]. Indeed, EGFR targeted therapy was found to increase overall survival by 10–20% in colorectal cancer [4]. However, resistance to this therapy was inevitable with the emergence of KRAS and BRAF mutations, driven by intrinsic and extrinsic mechanisms affecting both cellular pathways and tumor microenvironment, respectively.

Numerous studies have been conducted with the aim to overcome this resistance and improve the prognosis. One of the proposed investigational therapies is high dose vitamin C (intravenous route). Vitamin C, or ascorbic acid, is a water-soluble vitamin essential for humans, though they lack the ability to synthesize it and rely on its intake from diet [5,6]. Its main biologic function revolves around its ability to provide reducing equivalents, and therefore is a cofactor in several reactions that require reduction of iron or copper metalloenzymes. Due to its redox potential and involvement in several cellular processes, vitamin C is being investigated in clinical trials as a treatment modality for several types of cancers. In particular, cells harboring KRAS and BRAF mutations overexpress Glucose Transporter 1 (GLUT1) and sodium-dependent vitamin C transporter 2 (SVCT2), which leads to increased intracellular vitamin C transport, resulting in reactive oxygen species (ROS) generation and finally cell death. Therefore, high dose vitamin C is proposed to overcome resistance to EGFR targeted therapy in BRAF/KRAS mutated CRC. Furthermore, vitamin C exerts its anti-neoplastic effect via different mechanisms; it interferes with anaerobic glycolysis, which is the main metabolic pathway in cancer cells, and also hinders angiogenesis [7,8].

The aim of this review is to summarize the data available on the therapeutic use of high dose vitamin C in metastatic CRC (mCRC), in addition to expanding on its potential use in overcoming anti-EGFR resistance.

## 2. Methods and Search Strategy

A comprehensive search strategy was adopted using Pubmed and MEDLINE databases (up to 11 August 2022). The MeSH terms used for the search included “Vitamin C”, “Colorectal Neoplasm”, “KRAS mutation”, “EGFR resistance”, and “Ascorbic Acid”. We screened the abstracts to identify relevant articles, and we included basic science studies, clinical trials, and reviews on the topic. In addition, we screened the bibliography of the selected articles to identify important studies that could have been missed in the search. The included studies’ abstracts were checked for their relevance to the research question, and the eligible articles were extensively assessed for inclusion in this work. Additional papers were obtained for the bibliographies of the included studies. A thorough evaluation was conducted on the studies that have been chosen regarding their relation to the topic, results, and outcome.

## 3. Findings

### 3.1. Mechanisms of EGFR Resistance in mCRC

Although mCRC is associated with a very poor prognosis, almost half of the patients newly diagnosed with CRC first present in the metastatic phase. Combination chemotherapy has been regarded as the cornerstone for mCRC management [9]. Furthermore, anti-EGFR monoclonal antibodies (mAbs), such as Cetuximab, have proven to be an effective clinical therapy for mCRC patients with wild type KRAS tumors, and prolonged survival for 10–20% of patients [4]. One particular trial (CRYSTAL), revealed that the application of cetuximab and FOLFIRI can increase overall survival (OS) by 8.2 months and reduce the risk of progression by 15% in patients with KRAS wild type mCRC compared to FOLFIRI alone [10]. However, its clinical application is still limited due to high rates of drug resistance, such that treatment benefit has been shown to last only 8–10 months.

Numerous therapeutic strategies have been conducted and investigated to overcome the resistance to anti-EGFR mAbs. However, we must first understand the two mechanisms of resistance.

Intrinsic mechanisms include the activation of RAS/RAF/MEK/ERK and PI3K/AKT/mTOR cascades through genomic alterations and protein phosphorylation (as shown in Figure 1). Furthermore, compensatory feedback loop signaling of EGFR is stimulated by ERBB2/MET amplification and abnormal IGF-1R activation. In addition, epithelial-to-mesenchymal transition, glycolysis, lipid synthesis, fatty acid oxidation, and vitamin deficiency are also contributors to resistance [11].

On the other hand, the tumor microenvironment also plays a role in conferring extrinsic resistance to anti-EGFR therapy. This can include dysfunction of natural killer (NK) cells and macrophages that decrease the anti-EGFR antibody-dependent cellular cytotoxicity, and decreased density of effector T-cells and increased PD-L1 expression which assist cancer survival. Other factors leading to drug resistance are cancer-associated fibroblasts secreting mitogenic growth factors that activate RAS or MET pathways, in addition to abnormal angiogenesis [11].

Another aspect leading to acquired resistance in mCRC is caused by the emergence of a heterogeneous resistant population of surviving clones (i.e., persister cells, drug-sensitive RAS/BRAF wild type cells which were not eliminated by anti-EGFR targeted therapies) characterized by a drug tolerant state instigated by prolonged drug exposure, relying on different mechanisms, either genetic or non-genetic [12].

### 3.2. The Role of High Dose Vitamin C in Cancer

#### 3.2.1. Vitamin C Bioavailability and Requirements

Oral vitamin C produces tissue and plasma concentrations that the body tightly controls [13,14]. Normally, total body content of vitamin C ranges from 300 mg (at near scurvy) to about 2 g [14,15]. Plasma concentration of vitamin C is tightly controlled, and generally does not surpass 100 μM. Roughly, at moderate intakes of 30–180 mg/day, 70–90% of vitamin C is absorbed [15]. However, at doses above 1 g/day, absorption decreases by less than 50% and absorbed ascorbic acid is excreted in the urine [14,16]. However, when vitamin C is administered by IV route, it bypasses the gastrointestinal regulation and attains a dose-dependent plasma concentration. Nonetheless, its half-life is less than 2 h [17].

#### 3.2.2. High Dose Vitamin C in Cancer Clinical Trials

In 1979, Cameron, Pauling et al., showcased that patients with terminal cancer treated with high pharmacological doses of vitamin C (10 g/day by IV infusion for about 10 days and orally thereafter) had significantly prolonged survival rates and improved quality of life compared to matched controls that did not receive vitamin C [18]. In a randomized double-blinded placebo control study conducted at Mayo Clinic, high dose oral vitamin C did not replicate this favorable response in advanced cancer [19,20]. One possible explanation is that oral vitamin C does not achieve the required plasma concentration to exert an anti-cancer effect, unlike IV. Indeed, orally administered vitamin C achieves maximum plasma concentrations of no more than 220 μmol/L of blood, while high-dose IV vitamin C generates plasma concentrations up into the millimolar range (≥15 mmol/L), leading to different outcomes [16,21,22]. Yeom et al., evaluated the quality of life of 39 terminal cancer patients who received high dose vitamin C (10 g twice by IV and 4 g oral daily intake for a week). Patients reported significant improvement on the functional scale in terms of physical, role, emotional, and cognitive function, in addition to significant lower scores for fatigue, nausea/vomiting, pain, and loss of appetite [23]. Similar results were also found in other trials where high dose vitamin C was found to ameliorate quality of life for patients with terminal cancer on palliative care [24,25,26,27].

In addition, several clinical trials were conducted to test the safety of high dose vitamin C as monotherapy or in combination with other chemotherapeutic agents, as well as to determine the maximal tolerable dose that could be used in phase II and III trials. For instance, Wang et al., administered vitamin C with FOLFOX or FOLFIRI with or without bevacizumab to patients with colorectal or gastric cancer. The study reported no drug limiting toxicity and showed decreased hematological and gastrointestinal toxicities compared to other trials employing the same chemotherapy regimens for these two cancers [28,29,30,31,32]. Monti et al., administered vitamin C to 14 stage IV patients with pancreatic cancer receiving gemcitabine and erlotinib. In this study, eight patients had a decrease in the tumor size, seven had a stable disease and two had disease progression, although progression free survival and overall survival were comparable to those on gemcitabine/erlotinib alone [33]. In two other studies also on pancreatic cancer, vitamin C was found to decrease the rate of severe toxicity in patients receiving gemcitabine 500 mg/m, irinotecan 80 mg/m, leucovorin 300 mg, 5-fluorouracil (5-FU) 400 mg/m, and gemcitabine monotherapy [34,35]. Additional studies investigating the efficacy of vitamin C in advanced cancers did not show objective tumor response [26,27]. With respect to toxicity, mild side effects were recorded in some trials, mainly due to the osmotic load of the vitamin C infusion, and were reversible with adequate hydration [26,33], and one study reported kidney stone formation and hypokalemia as possibly related to vitamin C [36]. More severe side effects in these trials were related to the administration of chemotherapy. Furthermore, caution should be made in patients with G6PD deficiency, as high dose vitamin C can induce hemolysis, and patients should be screened before administration of IV vitamin C [37]. Results from several studies demonstrated that the optimal dose of IV vitamin C that could be adopted in phase II trials was 1.5 g/kg or 70 to 80 g/m^2^ [32].

#### 3.2.3. Role of Vitamin C in KRAS and BRAF Mutated Colorectal Cancer

RAS mutations are present in around 40% of mCRC, while BRAF mutations account for 10% [38]. Both mutations have been used as predictors of resistance to EGFR targeting drugs. In fact, testing for these mutations on tissue specimen of mCRC patients before the initiation of anti-EGFR therapy has become mandatory, particularly since resistance may be present originally or even may develop during the treatment in initially wild type patients; a phenomenon known as acquired (or secondary) resistance [39]. In order to identify these mutations, tissue and liquid biopsy method can be used as an analytical technique to detect tumor-derived biomarkers in body fluids such as circulating tumor DNA (ctDNA) [40]. Indeed, the detection of ctDNA released by cancer cells provides valuable information in relation to the prognosis and prediction of therapeutic resistance or sensitivity. Undeniably, improving detection of KRAS/BRAF mutations at different time points, enhancing correlation between its levels and survival and monitoring its response to therapy.

Furthermore, KRAS and BRAF mutations correlate with GLUT1 overexpression by cancer cells and excessive dependence on aerobic glycolysis as an energy source [41]. Aerobic glycolysis, also known as the Warburg effect, is a hallmark of cancer, in which glucose is converted to lactate despite the availability of oxygen. This is because pyruvate, the end product of glycolysis, is diverted from the mitochondria as a result of transcriptional activation of pyruvate dehydrogenase kinase 1, which in turn inactivates pyruvate dehydrogenase. As a result, the conversion of pyruvate to acetyl-CoA is hindered, and pyruvate is diverted into the cytosol where it is converted to lactate [42]. Although the shift from oxidative phosphorylation to aerobic glycolysis may seem to generate less energy per mol of glucose, the latter process is around 10 to 100 times faster and thus generates more ATP per unit time compared to oxidative phosphorylation [43,44]. Even though aerobic glycolysis is not exclusive to cancer cells and occurs in normal rapidly growing cells [45,46], its activation is enhanced and sustained in cancer cells because of activation of oncogenes and loss of tumor suppressor genes [43]. In addition, aerobic glycolysis sustains the production of metabolic intermediates (carbon moieties) for the synthesis of cellular components of the growing tumor [15], producing reducing equivalents when these intermediates are shunted into the pentose phosphate pathway (PPP) and reducing the production of reactive oxygen species (ROS), which enhances cellular proliferation [14]. Finally, the accumulation of lactic acid renders the tumor microenvironment more acidic, which in turn drives genetic instability, favors tumor invasion, cell motility, epithelial-to-mesenchymal transition, metastasis, resistance to apoptosis, immune evasion, and enhances angiogenesis [13,14].

Therefore, targeting this rewired glucose metabolism can be an effective therapeutic option for KRAS and BRAF-mutant CRC. GLUT1 and GLUT3 transport the oxidized form of vitamin C, dehydroascorbate (DHA) into the cells, where it is reduced back to vitamin C at the expense of glutathione (GSH), thioredoxin, and NADPH. In their study, Yun et al., demonstrated that DHA transport was increased in KRAS and BRAF-mutant cells, and this was mediated by GLUT1 overexpression. The rapid uptake of DHA and its intra-cellular reduction to vitamin C depletes the reserves of glutathione, leading to ROS accumulation and GADPH inactivation (as shown in Figure 2). The end result is an energy crisis and apoptosis in KRAS and BRAF mutated cells, which is not observed in wild-type CRC cells [42]. Aguilera et al., also demonstrated the vitamin C induced disruption of the Warburg metabolism in KRAS mutant CRC cells. Intracellular vitamin C causes detachment of RAS from the plasma membrane, thereby blocking the phosphorylation of PKM2 (pyruvate kinase M2), leading to downregulation of GLUT-1 expression [43,44].

In a study conducted by Jung et al., L-ascorbic acid induced cell death when partnered with cetuximab. This was mainly demonstrated in human colon cancer cells with a mutant KRAS gene, influenced by sodium-dependent vitamin C transporter 2 (SVCT-2) with administration of daily doses of 10 g of L-ascorbic acid for 6 h. Specifically, the knockdown of endogenous SVCT-2 induced resistance to L-ascorbic acid treatment in SVCT-2-positive cells, whereas ectopic expression of SVCT-2 induced sensitivity to L-ascorbic acid treatment in human CRCs that do not express SVCT-2 [45]. In addition, differences in SVCT-2 expression revealed a clear correlation with sensitivities to cetuximab and L-ascorbic acid (as shown in Figure 2). Particularly, recent studies that showed flow of L-ascorbic acid into the cell via SVCT-2 but not SVCT-1 support these findings. Taken together, these outcomes suggest that SVCT-2 expression may enable bypassing resistance to cetuximab in human colon cancer patients with a mutant KRAS by L-ascorbic acid.

Another approach to overcoming secondary resistance to EGFR blockade is targeting cellular proliferation axes with a variety of drugs. At first, MAPK signaling pathway was assessed; particularly over the past 30 years, research has proven that it plays a crucial role in initiating a wide range of cellular responses (proliferation, migration, differentiation, and apoptosis) by converting extracellular stimuli [46]. However, since no significant changes were noted when evaluating the MAPK signaling pathway upon the addition of cetuximab, focus was shifted towards changes in the RAF-MEK-ERK pathway after L-ascorbic acid exposure. Remarkably, decreased phospho-MEK, phospho-ERK, phospho-BRAF, and phospho-CRAF were noted when both drugs were used in the treatment [45], all of which are known to be key molecules for EGFR resistance in mutant KRAS human CRC cells expressing SVCT-2.

CRAF has been known to bind to ASK-1, suppressing its pro-apoptotic activity. According to their study, Jung et al., revealed that activation of ASK-1 and p38 pathway was induced by L-ascorbic acid and cetuximab in SVCT-2 expressing cells [47]. These findings suggest that SVCT-2-dependent reactive oxygen species production induces the activation of the ASK-1 p38 pathway, modulating cellular apoptosis. Changes in these signaling molecules were observed only in the tissues from the mutant KRAS and SVCT-2-positive human colon cancer cell line SW620 but not in tissues from the mutant KRAS and SVCT-2-negative cell line HCT116.

While long-term and larger studies are still lacking, available data supports the notion that L-ascorbic acid overcomes resistance to cetuximab by initiating the ASK-1-mediated apoptosis pathway through the blockade of the MAPK signaling pathway.

Although not enough clinical evidence favors the use of high dose vitamin C in KRAS or BRAF mutated CRC, data from a phase III clinical trial revealed promising results. A total of 442 patients were assigned to receive either chemotherapy (control group) or chemotherapy plus high-dose IV vitamin C 1.5 g/kg/day on day 1–3 (experimental group) and were followed up for 24.5 months [48]. PFS, ORR, and OS were similar between the control and experimental group. At first, results revealed that chemotherapy alone yielded a superior PFS when compared to chemotherapy plus high dose vitamin C. However, the prespecified subgroup analysis of patients with RAS mutation showed improved PFS in the experimental group only in patients with mCRC and KRAS mutation (9.2 vs. 7.8 months, HR 0.67; 95% CI, 0.50–0.91; *p* = 0.01). Finally, treatment related adverse effect of grade 3 or higher occurred in 33.5% of the patients in the experimental group, as compared with 30.3% in the control group [48]. Table 1 showcases the results of various trials using vitamin C as treatment. 

## 4. Conclusions

Despite the lack of robust supportive trials, high dose vitamin C shows a promising role in overcoming EGFR resistance in mCRC with wild KRAS mutation with resistance to anti-epidermal growth factor inhibitors and in patients with KRAS and BRAF mutations; specifically with mutant cells overexpressing GLUT1 and SVCT2, both of which enable intracellular vitamin C transport, leading to reactive oxygen species generation and finally cell death.

Furthermore, while this experiment measured vitamin C efficiency, additional larger prospective trials are required to consolidate this finding and gain more insight on anti-EGFR resistance mechanisms; to investigate, for instance, development of effective therapies, with promising second-generation antibodies and combinations with MET signaling pathway, or MEK inhibitors and pan-ERBB.

## Figures and Tables

**Figure 1 biomedicines-11-00678-f001:**
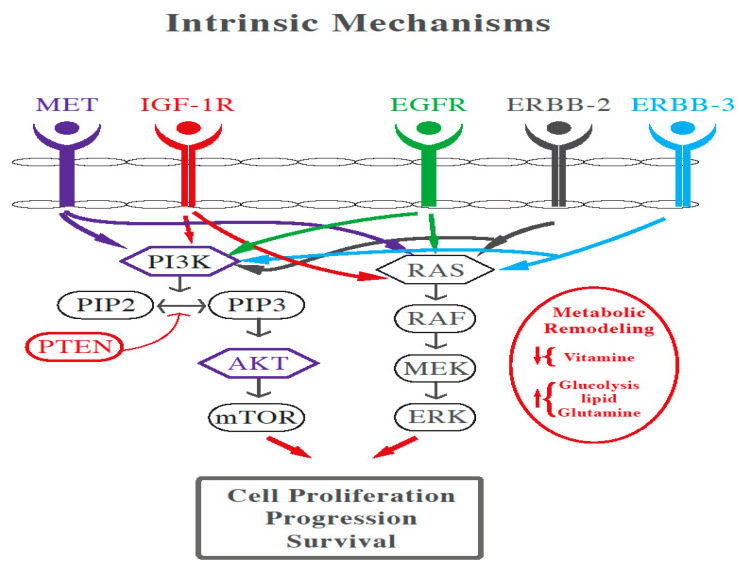
Cross talk between MAPK, PI3K, and Wnt pathway in CRC.

**Figure 2 biomedicines-11-00678-f002:**
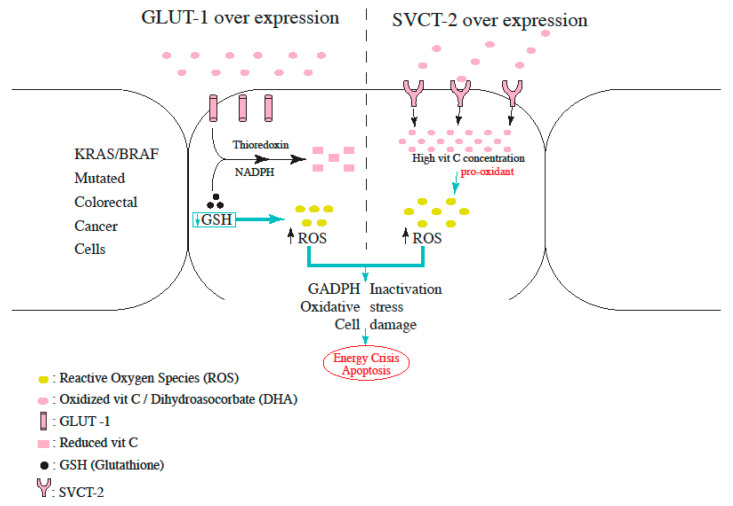
Proposed mechanism for increased ROS production in L-Ascorbic Acid-treated cells overexpressing either GLUT-1 or SVCT-2.

**Table 1 biomedicines-11-00678-t001:** Results of various trials using vitamin C as treatment.

Study	Design and Population	Aim	Intervention	Results	Notes
**Wang (2019) NCT02969681** [25]	-Phase I open label single center dose escalation speed expansion -30 mCRC and 6 mGC	Determine maximum tolerated dose of AA w/mFOLFOX or FOLFIRI +/− bevacizumab	Part 1 (dose escalation): AA (0.4, 0.6, 0.8, 1, 1.2 and 1.5 g/kg) on days 1 to 3 of FOLFOX or FOLFIRI every 14 d. Part 2 (dose expansion): AA given 1.5 g/kg or MTD on days 1 to 3. Tx duration: 12 cycles or progression or side effects.	No DLT in part 1 or part 2 and MTD not reached → 1.5 g/kg chosen as RP2D. Disease control rate 95.5%. No difference in efficacy between wt and m KRAS/BRAF CRC. Median PFS of the entire cohort 8.8 m.	The current study showed markedly decreased all-grade and grade ≥ 3 bone marrow and gastrointestinal toxic effects compared with previous trials investigating the same chemotherapeutic regimens in mCRC or mGC.
**Monti 2012** [26]	-Phase I open label dose escalation -A total of 14 stage IV pancreatic cancer patients receiving gemcitabine and erlotinib (not previously treated)	Primary: safety Secondary: response to tx	First cohort received 50 g IV AA per infusion, second cohort received 75 g/infusion, and third cohort received 100 g/infusion. A cycle consisted of three infusions per week performed on separate days, for 8 weeks.	A total of 9 patients completed the study. Side effects: mild headache and nausea from osmotic load that resolve; 8 serious adverse events recorded but related to gemcitabine/disease progression; 8 patients had dec in tumor size, 7 patients had stable disease, and 2 progressed.	Med PFS 89 days and med OS 182 d (comparable to gem/erlo alone).
**Bruckner 2017 (abstract)** [27]	-Phase II trial, open label -A total of 26 patients with advanced pancreatic cancer		High dose AA (75–100 g) 1–2 x/week with GFLIP Q2w until progression.	Decreased rate of severe toxicity.	
**Welsh (2013)** [28]	-Phase I single institution, prospective, open label -A total of 9 patients with stage IV pancreatic cancer receiving gemcitabine	Safety and tolerability of AA with gemcitabine	Twice weekly (50–125 g) IV AA and concurrent gemcitabine until DLT or progression. Target peak AA level > 350 mg/dl.	A total of 6/9 patients maintained/improved PS. PFS 26 +/− 7 weeks and OS 12 m. Adverse events related to AA were rare and included diarrhea and dry mouth. Adverse events were less severe when compared to published data for gemcitabine alone.	
**Stephenson** [20]	-Phase I, single center, non-comparative dose escalation -A total of 17 patients with advanced cancers not responsive to standard tx	Safety and tolerability of pharmacokinetics of high dose IV AA as monotherapy in advanced tumors	A total of 5 cohorts of 3 patients receiving dose escalation (30 g/m2 and inc by 20) until MTD	No objective tumor response. Side effects were mild and possibly related to treatment. Some patients had improved qol score at 3 and 4 weeks.	Dose of 70 to 80 g/m2 appears to be optimal for future studies.
**Hoffer (2008)** [19]	-Phase I, single center, dose escalating. -A total of 24 patients with advanced cancers, pretreated. They did not receive chemo with AA.	Document the safety and clinical consequences of i.v. ascorbic acid administrated in a dose sufficient to sustain plasma ascorbic acid concentrations >10 mmol/l for several hours	Cohorts receiving fixed doses of 0.4, 0.6, 0.9, and 1.5 g/kg for 4 weeks cycle	Mild clinical toxicity occurred, all consistent with the SE attending the rapid infusion of any high-osmolarity solution. Preventable by encouraging patients to drink fluids. No objective tumor response, but 2 patients in the 0.6 group had stable disease. AA could be promising when combined with cytotoxic agents.	1.5 g/kg (infused > 90–120 min 3 x/w) was adopted as the recommend dose for future phase II trials
**Riordan** [29]	-Pilot study -24 late stage terminal cancer patients	Clinical safety of high dose AA	Continuous infusions of 150 to 710 mg/kg/day for up to eight weeks	Most SE were mild and 2 were grade 3 possibly related to AA: kidney stone and hypoK. One patient had stable disease and continued the treatment for 48 weeks. AA is relatively safe, provided the patient does not have a history of kidney stone.	
**Sartore-Bianchi (2022)** [49]	-Open-label, single-arm phase 2 clinical trial -A total of 52 patients with tissue-*RAS* WT tumors after a previous treatment with anti-EGFR-based regimens underwent an interventional ctDNA-based screening.	Exploiting blood-based identification of *RAS*/*BRAF*/*EGFR* mutations levels to tailor a chemotherapy-free anti-EGFR rechallenge with panitumumab	A total of 36 patients were molecularly eligible for panitumumab rechallenge. Of these, 27 received the drug as per trial protocol, 6 did not meet clinical inclusion criteria, and 3 were treated otherwise as per physician choice	Of 27 enrolled patients, 8 (30%) achieved partial response and 17 (63%) disease control, including 2 unconfirmed responses. These clinical results favorably compare with standard third-line treatments and show that interventional liquid biopsies can be effectively and safely exploited in a timely manner to guide anti-EGFR rechallenge therapy with panitumumab in patients with mCRC.	
**Wang (2022)** [48]	-Randomized, open labeled, multicenter phase II -A total of 442 histologically confirmed mCRC patients with normal glucose-6-phosphate dehydrogenase status and no prior treatment for metastatic disease	Compare the efficacy and safety of high-dose vitamin C plus FOLFOX +/− bevacizumab versus FOLFOX +/− bevacizumab as first-line treatment in patients with metastatic colorectal cancer (mCRC)	A total of 442 patients were randomized into a control (FOLFOX +/− bevacizumab) and an experimental (high-dose vitamin C (1.5 g/kg/d, intravenously for 3 h from D1 to D3) plus FOLFOX +/− bevacizumab) group	In prespecified subgroup analyses, patients with RAS mutation had significantly longer Progression Free Survival (median PFS, 9.2 vs. 7.8 months; HR, 0.67; 95% CI, 0.50–0.91; *p* = 0.01) with vitamin C added to chemotherapy than with chemotherapy only.	

## Data Availability

Not applicable.

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
