# Peer review of "Overcoming EGFR Resistance in Metastatic Colorectal Cancer Using Vitamin C: A Review"

_biomedicines, 2023, doi:10.3390/biomedicines11030678_

Round 1

Reviewer 1 Report

Machmouchi et al. highlight the relevance of Vitamin C to overcome resistance in metastatic colorectal cancer. However, the topic has timeliness and importance in the current context of dietary intervention and cancer therapies including the reversal of drug resistance.

This manuscript has important points to be considered for better suitability to this journal.

1.      Abstract should show the rationale for linking Vitamin C and EGFR resistance. There are no obvious needs to include combinations of search words. Readers will understand the rationale and objectives of the manuscript. Hence, the abstract should be rewritten and restructured.

2.      Introduction lacks the coherence to support the rationale and title of this manuscript. An introduction should start with problems and needs in colorectal cancer, EGFR resistance and mechanisms, the role of Vitamin C in colorectal cancer and cancer specifically metastasis, then rationale to link Vitamin C and EGFR drug resistance, preclinical and clinical status on uses of Vitamin C, then last paragraph should emphasize what incremental discussion is offered in this manuscript.

3.      The flow and relevance of the subsequent section are not coherent. Hence, this manuscript needs restructuring and rewriting.

4.      A paragraph is important to highlight that loss of the GULO gene humans predisposes to humans with dietary sources of Vitamin C. Even the bioavailability of Vitamin C and cancer therapies should be discussed.

Author Response

Thank you for giving us the opportunity to submit out review paper entitled “Overcoming EGFR resistance in metastatic colorectal cancer using Vitamin C: a review” to biomedicines. We appreciate the time and effort that you have dedicated to providing your valuable feedback on our manuscript.

We have been able to incorporate changes to reflect most of the suggestions provided by the reviewers. Here is a point-by-point response to the reviewers’ comments and concerns.

  1. Abstract should show the rationale for linking Vitamin C and EGFR resistance. There are no obvious needs to include combinations of search words. Readers will understand the rationale and objectives of the manuscript. Hence, the abstract should be rewritten and restructured.

Point 1: Abstract is rewritten and restructured. We showcased the linking of Vitamin C and EGFR resistance. In addition, Combination of search words has been omitted.

  1. Introduction lacks the coherence to support the rationale and title of this manuscript. An introduction should start with problems and needs in colorectal cancer, EGFR resistance and mechanisms, the role of Vitamin C in colorectal cancer and cancer specifically metastasis, then rationale to link Vitamin C and EGFR drug resistance, preclinical and clinical status on uses of Vitamin C, then last paragraph should emphasize what incremental discussion is offered in this manuscript.

Point 2: The introduction has been rewritten and restructured so that the chain of thought would follow the proposed one.

  1. The flow and relevance of the subsequent section are not coherent. Hence, this manuscript needs restructuring and rewriting.

Point 3: Changes have been made to the sections (re-ordered) of the manuscript so that the flow and relevance of the subsequents sections is coherent.

  1. A paragraph is important to highlight that loss of the GULO gene humans predisposes to humans with dietary sources of Vitamin C. Even the bioavailability of Vitamin C and cancer therapies should be discussed.

Point 4: Although GULO gene plays a significant part in the ability to synthesize vitamin C (responsible for catalyzing the last step of vitamin C biosynthesis). However, we only broadly elaborated the angiogenesis process, Hence it may not be significant in our paper since the main objective of our review is to showcase overcoming EGFR resistance with available high dose Vitamin C. But we added some references on the topic.

We have added a paragraph on the bioavailability of Vitamin C.

We are grateful to the reviewers for their insightful comments on our paper.

Regards

Reviewer 2 Report

Dear Authors,

You present here a very interesting review of the use of vitamin C in overcoming EGFR resistance in metastatic colorectal cancer. 

There is no journal template used in this manuscript.

In the Abstract, there are 3 different fonts and letter size. 

Phrases "We searched...improved outcomes" should not be part of the Abstract.

What was the period of time investigated? You say only "up to August 11, 2022". Since when?

Please use the entire names, not only abbreviations (ex. PFS, OS, GFLIP, etc.).

Reduce the dimension of figures, they are way too large comparing to text.

You do not precise if there were side effects mentioned for the high-dose intake of vitamin C. Also, considering the important role played by vitamin C in the absorption of iron, did you find data regarding sideremia levels in these high-dose of vitamin C regimen?

The Conclusion part has to be detailed.

Author Response

Thank you for giving us the opportunity to submit out review paper entitled “Overcoming EGFR resistance in metastatic colorectal cancer using Vitamin C: a review” to biomedicines. We appreciate the time and effort that you have dedicated to providing your valuable feedback on our manuscript.

We have been able to incorporate changes to reflect most of the suggestions provided by the reviewers. Here is a point-by-point response to the reviewers’ comments and concerns.

  1. There is no journal template used in this manuscript.

Point 1: Manuscript template has been edited.

  1. In the Abstract, there are 3 different fonts and letter size. 

Point 2: Fonts and letter sizes have been adjusted.

  1. Phrases "We searched...improved outcomes" should not be part of the Abstract.

Point 3: Abstract is rewritten and restructured. In addition, Combination of search words has been omitted.

  1. What was the period of time investigated? You say only "up to August 11, 2022". Since when?

Point 4: We did not limit our investigation to a specific starting point. Hence, we were not able to provide a starting period of time.

  1. Please use the entire names, not only abbreviations (ex. PFS, OS, GFLIP, etc.).

Point 5: We have added the full names instead of the abbreviations.

  1. Reduce the dimension of figures, they are way too large comparing to text.

Point 6: Figure dimensions have been reduced

  1. You do not precise if there were side effects mentioned for the high-dose intake of vitamin C. Also, considering the important role played by vitamin C in the absorption of iron, did you find data regarding sideremia levels in these high-dose of vitamin C regimen?

Point 7: We added a part in section 3.2.2 highlighting the vit C side effects.

Regarding the effects of high dose vit C on iron absorption, according to Gerster et al “even high vitamin C intakes do not cause iron imbalance in healthy persons and probably in persons who are heterozygous for hemochromatosis”. (Please see below)

The question has been raised whether high-dose intakes of vitamin C might unduly enhance the absorption of dietary iron in persons with high iron stores or in patients with iron overload, possibly increasing the potential risk of iron toxicity. Extensive studies have shown that overall the uptake and storage of iron in humans is efficiently controlled by a network of regulatory mechanisms. Even high vitamin C intakes do not cause iron imbalance in healthy persons and probably in persons who are heterozygous for hemochromatosis. The uptake, renal tubular reabsorption and storage of vitamin C itself are also strictly limited after high-dose intake so that no excessive plasma and tissue concentrations of vitamin C are produced. [1]

  1. The Conclusion part has to be detailed:

Point 8: Conclusion has been amended.

  1. Gerster, H., High-dose vitamin C: a risk for persons with high iron stores? Int J Vitam Nutr Res, 1999. 69(2): p. 67-82.

We are grateful to the reviewers for their insightful comments on our paper.

Regards

Round 2

Reviewer 1 Report

The authors have considered the majority of suggestions to enhance the impact of the manuscript. 

Reviewer 2 Report

Dear Authors,

Thank you for considering my comments and suggestions.